Perineuronal satellite neuroglia in the telencephalon of New Caledonian crows and other Passeriformes: evidence of satellite glial cells in the central nervous system of healthy birds?

Medina Felipe S. 1 2
Hunt Gavin R. 1
Gray Russell D. 1
Wild J. Martin 2
Kubke M. Fabiana 2 f.kubke@auckland.ac.nz
1 School of Psychology, University of Auckland , New Zealand
2 Department of Anatomy with Radiology, University of Auckland , New Zealand
Bentley George
Electronic publication date: 2013 Jul 25
Publication date: 2013
Volume: 1
Electronic Location ID: e110
Received 2012 Nov 16; Accepted 2013 Jul 2
Copyright: © 2013 Medina et al.
Copyright year: 2013
Copyright holder: Medina et al.
License: This is an open access article distributed under the terms of the Creative Commons Attribution License, which permits unrestricted use, distribution, and reproduction in any medium, provided the original author and source are credited.
License URL: https://creativecommons.org/licenses/by/3.0/

Keywords: Perineuronal satellitosis, Oligodendroglia, Glia

Funding: The Marsden Fund, administered by the Royal Society of New Zealand Contracts 05-UOA-503 and 09-UOA-121 A Chilean CONICYT fellowship to Felipe Salvador Medina Rodriguez This work was funded by the Marsden Fund, administered by the Royal Society of New Zealand and a Chilean CONICYT fellowship to Felipe Salvador Medina Rodriguez. The funders had no role in study design, data collection and analysis, decision to publish, or preparation of the manuscript.

==============================
Glia have been implicated in a variety of functions in the central nervous system, including the control of the neuronal extracellular space, synaptic plasticity and transmission, development and adult neurogenesis. Perineuronal glia forming groups around neurons are associated with both normal and pathological nervous tissue. Recent studies have linked reduction in the number of perineuronal oligodendrocytes in the prefrontal cortex with human schizophrenia and other psychiatric disorders. Therefore, perineuronal glia may play a decisive role in homeostasis and normal activity of the human nervous system.

Here we report on the discovery of novel cell clusters in the telencephala of five healthy Passeriforme, one Psittaciform and one Charadriiforme bird species, which we refer to as Perineuronal Glial Clusters (PGCs). The aim of this study is to describe the structure and distribution of the PGCs in a number of avian species.

PGCs were identified with the use of standard histological procedures. Heterochromatin masses visible inside the nuclei of these satellite glia suggest that they may correspond to oligodendrocytes. PGCs were found in the brains of nine New Caledonian crows, two Japanese jungle crows, two Australian magpies, two Indian mynah, three zebra finches (all Passeriformes), one Southern lapwing (Charadriiformes) and one monk parakeet (Psittaciformes). Microscopic survey of the brain tissue suggests that the largest PGCs are located in the hyperpallium densocellulare and mesopallium. No clusters were found in brain sections from one Gruiform (purple swamphen), one Strigiform (barn owl), one Trochiliform (green-backed firecrown), one Falconiform (chimango caracara), one Columbiform (pigeon) and one Galliform (chick).

Our observations suggest that PGCs in Aves are brain region- and taxon-specific and that the presence of perineuronal glia in healthy human brains and the similar PGCs in avian gray matter is the result of convergent evolution. The discovery of PGCs in the zebra finch is of great importance because this species has the potential to become a robust animal model in which to study the function of neuron-glia interactions in healthy and diseased adult brains.

Introduction

Satellite glia were first described by Ramón y Cajal (1910) and Ramón y Cajal (1899) in healthy peripheral nervous tissue and in the 1930s the term perineuronal satellitosis (PS) was coined to describe neurons closely surrounded by multiple glia (see Vijayan et al., 1993). Today, most neuropathology textbooks teach us to recognise PS in the process of diagnosis of common pathologies that affect the nervous tissue (e.g., grade II astrocytoma, type I neurofibromatosis and anaplastic oligodendroglioma) (Haberland, 2007; Oemichen, Auer & König, 2006; Perry & Brat, 2010; Tonn, Westphal & Rutka, 2010). Satellite glia are, however, associated both with normal and pathological nervous tissue depending on the function and the type of perineuronal glia involved (Ludwin, 1984; Yokota et al., 2008; Faber-Zuschratter et al., 2009; Takasaki et al., 2010; Szuchet et al., 2011). Perineuronal (and polar) satellite glia have been described in healthy tissue in different regions of the human brain such as the cerebral cortex, hippocampus, basal ganglia and thalamus (Brownson, 1956; Vijayan et al., 1993; Vostrikov, Uranova & Orlovskaya, 2007). Satellite glia present in these brain areas mostly correspond to oligodendrocytes (Brownson, 1956; Vijayan et al., 1993; van Landeghem, Weiss & von Deimling, 2007; Vostrikov, Uranova & Orlovskaya, 2007; Kim & Webster, 2010; Kim & Webster, 2011; Takasaki et al., 2010). In addition, a reduction in the number of perineuronal oligodendrocytes (pN-OLG) in the human prefrontal cortex has recently been linked to schizophrenia and other psychiatric disorders (Vostrikov, Uranova & Orlovskaya, 2007; Kim & Webster, 2010; Kim & Webster, 2011).

One recent study has identified that gray matter pN-OLGs in both rats and humans are of the non-myelinating phenotype (Szuchet et al., 2011). In mice, evidence also suggests that these pN-OLGs in the somatosensory cortex support neuronal survival, differentiation, and function, and that they provide protection against neuronal apoptosis but do not play a major role in the plasmalemmal uptake of extracellular glutamate neurotransmitter (Takasaki et al., 2010). In contrast, evidence from one human study (van Landeghem, Weiss & von Deimling, 2007) shows that pN-OLGs in the neocortex and hippocampus express specific (PACAP) neuropeptides which are known to (1) both promote proliferation and retard maturation and myelogenesis in oligodendrocyte progenitors (Lee et al., 2001; Lelievre et al., 2006) and (2) modulate neuronal synaptic strength (Kondo et al., 1997). Human pN-OLGs also express region-specific glutamate transporters, which suggests they might help clear extracellular glutamate and thus prevent delayed neuronal and glial death in hypoxia-sensitive regions following transient global human ischemia (van Landeghem, Weiss & von Deimling, 2007). Furthermore, pN-OLG constitute a recently identified lineage of oligodendrocytes, one with a phenotype that “...blur[s] the boundary between a neuron and a glial cell...” (Szuchet et al., 2011). Taken together, these findings suggest that perineuronal glial satellites may play an instrumental role in the homeostasis and normal activity of the nervous system involving cognitive functions (see Szuchet et al., 2011).

Here we report on the presence of perineuronal glial clusters (PGCs) in the brains of New Caledonian crows (NC crows) that were found in the process of other anatomical investigations (Medina, 2013). Although neural clusters have been previously described in Field-L of the avian auditory nidopallium (see Fortune & Margoliash, 1992) the PGCs we found are clearly different structures. A survey based on perikaryal stains revealed that PGCs were also present in at least four other passerine species (including zebra finches) and in two of the eight non-passerine species that we investigated (one charadriidae and one psittacid). To our knowledge, this is the first formal description of such structures in non-mammalian brains.

Materials and Methods

The work involving animals and the transport of material was done in compliance with and with the approval from the University of Auckland Animal Ethics Committee (R602, R840 R469 and R425) in accordance with the Animal Welfare Act 1999 (New Zealand), the University of Auckland Code of Ethical Conduct for the Use of Animals for Teaching and Research and the The National Animal Ethics Advisory Committee (NAEAC) Good Practice Guide for the Use of Animals in Research, Testing and Teaching. Brains of NC crows were imported into New Zealand in compliance with New Zealand Ministry of Agriculture and Forestry (New Zealand) regulations, and with approval to collect and export the NC crow brains from the Customary Authority on Maré, New Caledonia dated 8 July 2008.

New Caledonian crow specimens

Nine New Caledonian crows (Corvus moneduloides) (six males and three females) captured on the island of Maré, New Caledonia, in August/October 2007 were used for this study. The birds were caught with a ‘whoosh net’ (obtained from SpiderTech Bird Nets, Helsinki, Finland). Female and male distinction was based on bill morphology and body weight on the day of capture (Kenward et al., 2004). Crows were housed in a 5-cage outdoor aviary situated in primary forest inland from the coast, for up to a maximum period of five months during which they participated in behavioural experiments (e.g., Taylor et al., 2009; Taylor et al., 2010; Medina et al., 2011). The cages were 3 m high and at least 4 m × 2 m in area. All cages were provided with ample perching space, branches and feeding logs. In addition to obtaining meat in experimental trials, the crows were fed a main meal in the evening consisting of soaked dog/cat biscuits (40+ mL of dry biscuits soaked in water for 15 min), bread or rice and occasionally raw egg. Papaya and clean drinking and bathing water were available throughout the day.

All crows were euthanised at the end of their captive period. Crows were caught and put into a clean, dark cloth bag and then injected with a lethal dose of pentobarbital (30 mg/kg, intramuscular Ludders, 2008) and held calmly in the dark bag by the experimenter until cardiac arrest. Immediately after cardiac arrest the crow was transcardially-perfused by gravity with 0.5 L of 0.9% saline followed by 0.5 L of 4% formaldehyde solution. The crow’s brain was then extracted from the skull and immersed in 4% formaldehyde for importation into New Zealand.

Tissue preparation

Brains were re-immersed and fixed in 4% paraformaldehyde (PFA) in 0.1 M sodium phosphate buffer (PB) upon arrival in New Zealand after perfusion and kept in cold storage. Brains were finally processed between 2008 and 2011. First, they were sectioned mid-sagitally and both hemispheres were cryoprotected in 30% sucrose in phosphate buffered saline (PBS 0.01M) for 4 to 7 days (until they sank twice in fresh sucrose solution). Brain halves were then placed in a solution of 15% gelatine with 30% sucrose (cryoprotective gelatine solution) at 40°C for one hour. The hemispheres were then placed in a custom-made mould so that fiduciary points could be made in the gelatine for later alignment of tissue sections. The mould consisted of a custom made plastic box with a removable base. The base had small holes drilled in a 3 mm grid pattern, on top of which a perfectly flat 5 mm thick layer of cryoprotective gelatine solution was left to set for 10 min prior to hemisphere placement. The brains were placed on top of this cryoprotective gelatine base with the midline facing down and left to set for 3 min at 6°C. Then, seven to ten sewing pins were inserted into the holes that had been drilled in the base of the mould so that they surrounded the brain hemisphere. A different cryoprotective gelatine solution containing a teaspoon of black fabric dye (to darken the solution) was then poured over the brain. This coloured solution had been previously prepared and kept in liquid state just above room temperature. Once poured, it was left to set first at 6°C, and then at −4°C, for a total time of 15 min.

The resulting cryoprotective gelatine block (containing the brain) was then removed from the mould, trimmed and placed, along with the pins, into 4% PFA 30% sucrose overnight. The pins were then removed and the block was sectioned on a sliding freezing microtome at 50 µm thickness in the sagittal plane. Sections were collected in 0.01% sodium azide PBS solution. For each hemisphere every third section was mounted serially onto gelatine chrome-alum coated slides, stained with haematoxylin neutral red, neutral red or cresyl violet (see below), dehydrated and coverslipped with DPX mounting medium (Scharlau) from xylene. The remaining sections were set apart in two series, one for storage and the other for immunocytochemical staining (see below).

Sections and fiduciary points in the surrounding dyed gelatine were imaged using a Leica M205 FA stereomicroscope with a mounted Leica DFC 500 digital camera, and the images subsequently merged and flattened in Adobe Photoshop CS3 Extended and then loaded into Visage Imaging AMIRA 5.2.0 for alignment. The aligned image stacks were then labelled using CorelDRAW X5 for the production of a complete brain atlas for the species. Brain regions were identified using boundary lines that could be recognised from the histological staining in the image stack (Fig. 1). Boundaries were carefully identified with the aid of several avian brain atlases (Karten & Hodos, 1967; Kuenzel & Masson, 1988; Izawa & Watanabe, 2007) and named according to the Avian Consortium Nomenclature (Reiner et al., 2004; Jarvis et al., 2005).

Figure 1 Medial sagittal section through the right brain of an adult male NC crow stained with Cresyl Violet (top) and schematic of the same section showing regional boundaries (bottom).

Rostral is left, ventral is down. Scale bar: 2 mm.

Classical histology

Brain sections were mounted on gelatine subbed glass slides and subjected to different classical histological methods for histological analysis.

Neutral red nuclear staining method

Sections were first dehydrated then re-hydrated and quickly rinsed in tap water. They were then placed in neutral red solution (5 mg neutral red in 500 mL distilled water and 20 mL acetate buffer 0.037M, pH 4.8) for 2–4 min, briefly rinsed with tap water and finally differentiated in 95% ethanol, dehydrated and coverslipped from xylene.

Carazzi’s modified haematoxylin staining method

Sections were first dehydrated then rehydrated and quickly rinsed in tap water. They were then left in haematoxylin solution (5 g haematoxylin and 25 g aluminium potassium sulphate, 100 mg potassium sodium iodate, in 100 mL glycerol, 400 mL distilled water and either 20 mL/L glacial acetic acid or 5 mL/L HCl) for 30–40 min, blued in running warm tap water for 5 min, and finally differentiated in 1% HCl in 70% ethanol for 2–5 s, dehydrated and coverslipped from xylene.

Haematoxylin neutral red nuclear staining method (with Carazzi’s haematoxylin)

Sections were dehydrated then rehydrated and quickly rinsed in tap water. They were then left in haematoxylin solution for 30–40 min before being blued in running warm tap water for 5 min and differentiated in 95% ethanol. The sections were next left in neutral red solution for 2 min, rinsed in warm tap water and finally differentiated in 95% ethanol, dehydrated and coverslipped from xylene.

Fast cresyl violet staining method

Sections were dehydrated then rehydrated and quickly rinsed in tap water. They were then left in cresyl violet solution (60 mL of 10 g/L cresyl violet in 540 mL distilled water and 5 mL 10% acetic acid) for 2 min, briefly rinsed in warm tap water, and finally differentiated in 95% ethanol, dehydrated and coverslipped from xylene.

Other specimens

Stained sagittal brain sections from two Indian mynahs (Acridotheres tristis), two Australian magpies (Gymnorhina tibicen), one purple swamphen (Porphyrio porphyrio), three zebra finches (Taeniopygia guttata), one monk parakeet (Myopsitta monachus), one Southern lapwing (Vanellus chilensis), one green-backed firecrown (Sephanoides sephanoides), one chimango caracara (Milvago chimango), and three pigeons (Columba livia) were also examined. The mynahs, Australian magpies and pukeko were perfused post-mortem after being collected in the wild between 2005 and 2008 in the North Island of New Zealand. The brains of these three species were sectioned and stained with cresyl violet by Dr. Jeremy Corfield at the University of Auckland using the same methods described above. The sagittal brain sections of two zebra finches (35 µm thick, cresyl violet staining) were provided by Dr. Priscilla Logerot at the University of Auckland. Coronal brain sections of one monk parakeet, one southern lapwing, one green-backed firecrown, one chamingo caracara and three pigeons were provided by Dr. Jorge Mpodozis and Dr. Gonzalo Marín at the Universidad de Chile. High-resolution images of sagittal brain sections (50 µm thick, cresyl violet staining) of two Japanese jungle crows (Corvus macrorhynchos) were provided by Dr. Ei-Ichi Izawa (Keio University, Tokyo, Japan). High-resolution images of brain sections of the barn owl (Tyto alba) and the chick (Gallus gallus) were studied on the free website http://brainmaps.org.

Tissue analysis

Microphotographs of the tissue were taken from different regions of the telencephalon (labelled as in Fig. 1) using a Nikon Digital Sight DS- 5MC camera attached to an Eclipse 80i Nikon microscope. The microphotographs were then loaded into Adobe Photoshop CS3 for cropping and figure production.

Results and Discussion

Our microscopy survey based on perikaryal stains in five passerine species (including NC crows) revealed the presence of tightly-packed cell clusters that, to our knowledge, have not been formally described to date in the avian brain literature. The cell clusters were first identified in the mesopallium, consisting of one or two larger central cells with long projections (classified as neurons) surrounded by a tight and compact cluster of four or more cells with no visible or very short projections (we classified these cells as perineuronal glia) (Fig. 2). We refer to these multi-cell structures consisting of a neuron and associated perineuronal glia as perineuronal glia clusters (PGCs).

Figure 2 Microphotographs of PGCs in the NC crow telencephalon. Orange arrowheads indicate neurons within a PGC and green arrows indicate unclustered neurons. White arrowheads show perineuronal glia, and the asterisks indicate presence of blood vessels or blood cells.

(A) Medium-sized PGC in the hyperpallium (light haematoxylin stain). Inset shows a large neuron (N) with its dendritic projections (visible contour indicated by dashed black line) showing a round nucleus (solid black circle) with a darkly stained nucleolus (n) in its centre, surrounded by nine perineuronal glia (white dashed lines) (B) Large PGC in the mesopallium (haematoxylin neutral-red stain). Inset showing the visible contour of the central neuron (black dashed line) and of 14 surrounding glia (white dashed line), neurons devoid of a glial cluster is also seen (green arrowhead); (C) Neuron in the hyperpallium apicale devoid of perineuronal cluster (green arrowhead) and isolate glia (white arrowhead). (D) Neurons and glia in the area parahippocampalis (dark haematoxylin stain) with no indication of perineuronal clustering. (E) Neurons (green arrowhead) and oligodendrocytes (white arrowhead) showing no clustered arrangement found in area corticoidea dorsolateralis (cresyl violet stain); (F) Unclustered neurons and glia in the arcopallium (haematoxylin neutral-red stain); (G) Large PGC in the mesopallium (haematoxylin neutral-red stain).

Initially, we saw PGCs in cresyl violet stained tissue, and then confirmed their presence with three other stains (haematoxylin, neutral red and haematoxylin-neutral red) in multiple sections from different individuals. This allowed us to eliminate the possibility that the identified clusters were the result of a histological artifact. Notably, these structures were not visible in calcium-binding protein immunocytochemical preparations (Medina, 2013). Additionally, PGCs visibly differ from the neuron clusters described in the avian auditory nidopallium (see Fig. 6 in Fortune & Margoliash, 1992). However, PGCs do resemble the perineuronal satellitosis described previously in the human hippocampus, i.e., “... clustering of glial nuclei was observed along one border or around the periphery of several neuronal perikarya...” (see Vijayan et al., 1993).

A well differentiated haematoxylin stain followed by a rapid neutral red stain provided the clearest delineation of the different cell types and their distribution within the cluster. Based on our perikarya stains, neurons were distinguished from glia by the presence of dark, coarsely stained Nissl substance, a large pale nucleus with a distinct central nucleolus, and lightly stained proximal segments of dendritic processes (Fig. 2). In contrast, glial cells contained much less visibly stained endoplasmic reticulum, giving them a compact round or oval form with darker stained nuclei and often with multiple spots of condensed chromatin. Similar criteria have been used in the past to distinguish glia from neurons (e.g., Sherwood et al., 2006).

Gross distribution of cell clusters in the avian telencephalon

Once we had successfully identified PGCs in the mesopallium of NC crows we carried out a careful microscopic survey of other telencephalic areas in this and three other passerine species (Australian magpie, Indian mynah and zebra finch), as well as in six non-passerines (purple swamphen, order Gruiformes; monk parakeet, Psittaciformes; green-backed firecrown, Trochiliformes; pigeon, Columbiformes; chamingo caracara, Falconiformes; Southern lapwing, Charadriiformes) using cresyl violet stained brain sections (Figs. 2–4). We also examined high resolution images of the brains of the two Japanese jungle crows (Passeriformes) and identified PGCs in this species. Far from having a simple, homogeneous pattern of distribution (i.e., presence or absence) across the whole telencephalon, we found that the apparent number of PGCs and their size varied according to the brain region and species under study. Also, the increment in size of the PGCs appears to be a direct function of the number of glia surrounding the central neuron(s).

In the striatum mediale the high cell density prevented unambiguous identification of PGS clusters in most species, although we were able to identify a few PGCs in both Corvus species. With the exception of the nidopallium caudale in the NC crows, PGCs in the pallial subdivisions of the passerine telencephala were easily recognised because in these regions cells are more sparsely distributed. The PGCs were most conspicuous in the mesopallium (Fig. 3) and hyperpallium densocellulare of the five passerines (Fig. 4), where their numbers appeared to be higher. A non-systematic microscopic survey of additional brain tissue material from five non-passerine species also revealed PGCs in the mesopallium of one psittacid (monk parakeet) and one charadriid (southern lapwing) (Fig. 5). The largest PGCs appeared to be present in the densocellular subregion of the hyperpallium and the mesopallium (Figs. 3 and 4). These telencephalic areas have been shown to form a neural circuit critical for audiovisual imprinting and passive avoidance learning in chicks (Horn, 1981; Horn, 2004; Cipolla-Neto, Horn & McCabe, 1982; Nakamori et al., 2010; Town & McCabe, 2011) and for colour-discrimination learning in pigeons (Chaves & Hodos, 1997) (though neither species present PGCs).

Figure 3 Microphotographs of the mesopallium of six avian species. Black arrowheads indicate neurons with perineuronal glia (white arrowheads) and the green arrowheads indicate unclustered neurons. The red arrowhead shows a neuron cluster.

The question mark indicates a neuron apparently surrounded by perineuronal glia (when tissue prevented unambiguous identification of glia). (A) NC crow; (B) Japanese jungle crow; (C) Australian magpie; (D) Indian mynah; (E) zebra finch; (F) pukeko. Scale bar: 100 µm.

Figure 4 Microphotographs of the hyperpallium densocellulare of six avian species.

Black arrowheads indicate neurons perineuronal glia (white arrowheads) and the green indicate unclustered neurons. The red arrowhead show a neuron cluster. The question mark indicates a neuron apparently surrounded by perineuronal glia (when tissue prevented unambiguous identification of glia). (A) NC crow; (B) Japanese jungle crow; (C) Australian magpie; (D) Indian mynah; (E) zebra finch; (F) pukeko. Scale bar: 100 µm.

Figure 5 Microphotographs of PGCs in the mesopallium of one psittacid and one charadriid.

Orange arrowheads indicate neurons within a PGC and green arrows indicate unclustered neurons. White arrowheads show perineuronal glia. (A) monk parakeet; (B) Southern lapwing.

Morphological cell identification was more difficult in the pukeko tissue due to its poor quality. Nevertheless we found some clusters, but these appeared to be mainly composed of large neurons (Figs. 3F and 4F). Neuron clusters (as distinct from PGCs) were also detected in the NC crow nidopallium caudale and have been described elsewhere (Fortune & Margoliash, 1992). Finally, cell clusters appeared to be absent in the hippocampus, area parahippocampalis, area corticoidea dorsolateralis, arcopallium and entopallium. We found no PGCs in pigeons, one chamingo caracara and one green-backed firecrown (however, this survey was less extensive because of the limited number of provided brain sections per species). Also, when closely inspecting whole high-resolution sagittal images of perikarya stained brains of the barn owl (Tyto alba, Strigiformes) and the chick (Gallus gallus, Galliformes) (available at http://brainmaps.org), we failed to find evidence of PGCs. In addition, we did not find PGCs in the three pigeons, one green-backed firecrown and one chamingo caracara.

Together, our findings indicate that avian PGCs are both region- and taxon-specific, suggesting that their presence in healthy passerines and humans (as shown by Brownson, 1956; Vijayan et al., 1993; van Landeghem, Weiss & von Deimling, 2007; Vostrikov, Uranova & Orlovskaya, 2007) may be the result of convergent evolution.

It is surprising that the presence of PGCs has not been previously reported in avian species, perhaps because neuropathology textbooks focus on perineuronal satellitosis in relation to mammalian brain disease (Haberland, 2007; Oemichen, Auer & König, 2006; Perry & Brat, 2010; Tonn, Westphal & Rutka, 2010). For example, previous observations of such clusters in zebra finch brains were interpreted as pathological (JM Wild, unpublished observations). It seems highly unlikely that the selective presence of PGCs in the telencephala of the brains that we studied was associated with developed neuropathologies. This is because the brains that we studied came from birds in different geographical locations with no signs of illness having been reported before perfusion. Further, the NC crow specimens were the subject of several behavioural experiments in which they excelled (e.g., Medina et al., 2011; Medina, 2013; Taylor et al., 2009; Taylor et al., 2010). Rather, our comparative studies suggest that PGCs are present in non-pathological avian brains, consistent with evidence of perineuronal satellitosis in both ill and healthy rodents (for evidence of PS in both ill and healthy rodents, see Ludwin, 1984; Krinke et al., 2000; Szuchet et al., 2011) and humans (see Vijayan et al., 1993).

That the presence of PGCs is brain region- and taxon-specific in birds invites speculation about their general function. Verkhratsky (2010, p. 1) stated that the neuronal web is embedded into a glial syncytium and gives rise to the sophisticated neuronal–glial network in which “... both types of neural cells [work] in concert, ensuring amplification of brain computational power”. As PGCs appear to be characteristic of passerine brains where they commonly occur in associative forebrain regions (i.e., mesopallium and hyperpallium densocellulare), it is tempting to hypothesise that they may play an important role in the flexible behaviour and problem-solving abilities seen in the highly successful, larger-brained passerine group (Emery & Clayton, 2005; Taylor et al., 2009). Since Psittaciformes constitute another taxon with enlarged forebrains and similar reports of innovative behaviour (see Emery & Clayton, 2004; Emery, 2006), the presence of PGCs in the mesopallium of the monk parakeet seems to support this view. However, PGCs were also found in one Southern lapwing, but the cognitive abilities of Charadriiformes have not been studied to date. Therefore, whether or not PGCs are linked to the emergence of higher cognitive abilities in Neornithes remains to be tested.

Conclusions

Our findings add to the increasing evidence suggesting that perineuronal glia (especially, pN-OLG) may play a major role in normal brain functioning. Future work, aimed at characterising in detail the biochemical and genetic signature of clustered avian perineuronal glia herein described, will be necessary. In particular, the visible nuclear chromatin masses in the perineuronal glia of NC crow brains suggest that the cells may correspond to oligodendrocytes (see also Mori & Hama, 1971; Ling et al., 1973; Uranova et al., 2001; Faber-Zuschratter et al., 2009). This suggests that the avian PGC may have a composition similar to that of mammalian perineuronal oligodendrocytes (pN-OLG), which if confirmed, would be of great importance. Thus, the discovery of PGCs in birds and their possible similarity with pN-OLG opens new avenues to explore their role in normal and diseased adult brains.

Most importantly, our discovery opens the way for comparative animal experimentation, which will help determine how PGCs may influence normal neuronal function. Our finding of PGCs in avian brains that closely resemble those previously described in the human hippocampus (see Vijayan et al., 1993) provides us with alternative animal models in which to explore the function and possible roles of PGCs in pathological conditions and cognition. The zebra finch provides a highly suitable animal model for comparative and functional studies to investigate the physiological function and possible role of PGCs in behaviour. Future biochemical, comparative and functional studies in Neornithes may elucidate whether PGCs have evolved in association with the emergence of complex cognition and/or specific audiovisual learning skills in members of this large avian group. Abbreviation list

A Arcopallium

B Nucleus basalis

Cb Cerebellum

CDL Area corticoidea dorsolateralis

CIO Capsula interna occipitalis

DLA Nucleus dorsolateralis anterior thalami

DLP Nucleus dorsolateralis posterior thalami

FPL Fasciculus prosencephali lateralis

FRL Formatio reticularis mesencephali pars lateralis

GP Globus pallidus

HA Hyperpallium apicale

HD Hyperpallium densocellulare

HVC High vocal centre

ICo Nucleus intercollicularis

Imc Nucleus isthmi pars magnocellularis

Imp Nucleus isthmi pars parvocellularis

LM Nucleus lentiformis mesencephali

M Mesopallium

MAN Nucleus magnocellularis nidopallii anterioris

MLd Nucleus mesencephalicus lateralis pars dorsalis

MSt Striatum mediale

N Nidopallium

NC Nidopallium caudale

SAC Stratum album centrale

TeO Tectum opticum

TnA Nucleus taeniae amygdalae

VeM Nucleus vestibularis medialis

X Area X

We would like to thank Dr. Jeremy Corfield for technical assistance and useful feedback during the initial development of the project and for providing some of the material used for this study. We would also like to thank Dr. Priscilla Logerot for providing the zebra finch brain material, and to Dr. Ei-Ichi Izawa for the high-resolution images of two Japanese jungle crow brains. We are also grateful to Dr. Jorge Mpodozis and Dr. Gonzalo Marín for providing brain tissue material of three pigeons, one green-backed firecrown, one chamingo caracara, one monk parakeet and one Southern lapwing. We are grateful to Dr Arie Perrie (UCSF) for looking at our material. We would also like to thank Alex Taylor and Mick Sibley for assistance in the field in New Caledonia.

Additional Information and Declarations

Competing Interests

Author Contributions

Ethics

Field Study Permissions

Data Deposition

M. Fabiana Kubke is an Academic Editor for PeerJ. The authors declare there are no other competing interests.

Felipe S. Medina Rodriguez conceived and designed the experiments, performed the experiments, analyzed the data, contributed reagents/materials/analysis tools, wrote the paper.

Gavin R. Hunt and Russell D. Gray conceived and designed the experiments, contributed reagents/materials/analysis tools, wrote the paper, provided input for the experimental work.

J. Martin Wild and M. Fabiana Kubke conceived and designed the experiments, contributed reagents/materials/analysis tools, wrote the paper, provided input for the experimental work, assisted data analysis.

The following information was supplied relating to ethical approvals (i.e., approving body and any reference numbers):

Import of brains into New Zealand was done under the standard guidelines of New Zealand MAF authority - (no special permit required). Permission to export of New Caledonian crow brains was issued by the Customary Authority on Maré, New Caledonia dated 8 July 2008.

The following information was supplied relating to ethical approvals (i.e., approving body and any reference numbers):

The work involving animals and the transport of material was done in compliance with and with the approval from the University of Auckland Animal Ethics Committee (R602, R840 R469 and R425) in accordance with the Animal Welfare Act 1999 (New Zealand), the University of Auckland Code of Ethical Conduct for the Use of Animals for Teaching and Research and The National Animal Ethics Advisory Committee (NAEAC) Good Practice Guide for the Use of Animals in Research, Testing and Teaching. The brains of the NC crows were imported into New Zealand in compliance with New Zealand Ministry of Agriculture and Forestry (New Zealand) regulations, and with approval to collect and export the NC crow brains from the Customary Authority on Maré, New Caledonia in a letter dated 8 July 2008.

The following information was supplied regarding the deposition of related data:

This work is part of Felipe Salvador Medina Rodriguez’s PhD thesis that will be deposited in the University of Auckland Institutional repository once the degree is granted. The original thesis chapter contains an expanded set of figures.

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
