# Peer review of "Perineuronal satellite neuroglia in the telencephalon of New Caledonian crows and other Passeriformes: evidence of satellite glial cells in the central nervous system of healthy birds?"

_PeerJ, doi:10.7717/peerj.110_

## Round 0.1 · original submission · Major Revisions

I find the comments of Reviewer 1 to be particularly constructive. I recommend that you read them in a positive light and follow them closely.

·

Basic reporting

This is a descriptive paper where the authors document the presence of perineuronal neuroglial cells in specific brain areas of healthy Passeriform bird species. They further claim that these cells are perineuronal oligodendrocytes (pN-OLGs) without providing unambiguous evidence for the identification. Throughout the manuscript the authors refer to these cells as pN-OLGs or “perineuronal neuroglia” as though they were one and the same, ignoring, thereby, the existence of other neuroglial cells such as astrocytes or microglial cells that may also function as perineuronal. The data as presented only support one of the claims: the existence of perineuronal cells in certain birds.

Experimental design

Whereas the authors used acceptable protocols to identify neurons and provide images that the readers can see and judge, they resorted to long passé histological staining to identify the satellite cells. Two criteria were used to define these cells as OLGs: one, condensed nuclear chromatin; and two, the opinion of a neuropathologist. None of these can be accepted as a scientific validation. That the authors should consider these as “successful criteria to identify pN-OLGs” is puzzling! Moreover, the authors do not present a single image of the so-called OLGs with sufficient magnification for the reader to actually see the cells and form an independent opinion as to what type they may be.

Validity of the findings

The authors have not provided convincing evidence that the perineuronal cells are of an OLG lineage. They have two options to: one, write a short report documenting the finding of perineuronal cells in specific areas of the bird’s brain without attempting to define the cells ; two, make a more substantial contribution by identifying these cells – preferentially - by immunohistochemistry. They could try using an anti-CXCR4 antibody to specifically identify pN-OLGs or antibodies against some of the other genes expressed by these cells ( Szuchet et al., 2011). If they fail to find antibodies that recognize avian cells and have no access to an electron microscope, and therefore must resort to a morphological identification, they should at least provide large enough images so the reader can assess their criteria for distinguishing between an oligodendrocyte, an astrocyte or a microglial cell. In mammals all three cell types can function as perineuronal.

Comments for the author

1. At least in mammals, pN-OLGs are of an OLG lineage but are phenotypically distinct from myelinating OLGs. It is, therefore, not appropriate to compare the morphology of pN-OLGs with the cells described by Mori and Leblond.
2. Do not use "satellitosis" to refer to normal perineuronal/satellite cells. The implication and definition of this term are "abnormal accumulation of satellite cells around an injured neuron.
3. The term "glia" (meaning glue) was introduced before the recognition that these were cells. It is time we stopped using the suffix "glia" as, e.g. oligodendroglia and switch to "cytes".
4. Check text for grammatical errors

·

Basic reporting

I read this paper with true pleasure - this is indeed a novel finding, well documented, original and interesting indeed. All in all commendable effort; I support publication of this paper in its present form.

Experimental design

Experimental design is adequate

Validity of the findings

The findings are original and sound. I have no criticism.

Comments for the author

Very nice paper; well presented well written and balanced.

---

## Round 0.2 · Major Revisions

Even though the reviewer suggested "minor" revisions, I consider the requested re-working of the manuscript to merit being classified as "major" revisions. Please take care to organize the manuscript as suggested, and give it to peers at your institution for comment prior to re-submission.

·

Basic reporting

This is a revised manuscript. The new version represents an improvement in one important aspect: the data support the claim. Nevertheless, the manuscript still needs to be revised.
Abstract: it reads like an introduction. Customarily no references are given in an abstract. If one or two must be included, the references should be given in full. Are the rules different for PeerJ?
There is an accepted sequence for an abstract: it starts with a couple of sentences of background, followed by a hypothesis or at least reasons for undertaking the studies, methods employed, results and significance. The background should be restricted to perineuronal cells. Consequently, the paragraph describing the functions of glial cells, in general, should be taken out. Perineuronal cells are independent entities; they have been characterized morphologically and ultrastructurally and – at least – perineuronal oligodendrocytes have been shown to be phenotypically distinct from myelinating glial cells. The same is probably true for the others. Nowhere do the authors state why they undertook this work. They could state that they were interested in finding out whether such cells were present in birds, particularly because neuronal clusters were previously described (Fortune & Margoliash, 1992). Only the most important findings should be listed and also their implication. Glial cells were defined as macroglial (i.e. astrocytes and oligodendrocytes) and microglial. It is redundant and incorrect to list macroglial, astrocytes, oligodendrocytes and microglial as the authors have done.
Introduction: this paper deals with perineuronal cells, therefore the focus should be on these cells and NOT on glial cells in general. The first paragraph is out of place. There is enough material in the literature to write a concise description of the current status of perineuronal cells, not limited to satellitosis.
Results and Discussion: The results are not presented in a logical sequence. They start by presenting the major findings, then they define the cells and only after do they give the criteria used for identification. This part should be reorganized.
Conclusions: should be a concise statement of major findings and their significance in a broader context. As written this section reads like a mixture of introduction and discussion. Additionally, there are paragraphs (e.g., the first one) that are not pertinent.
In sum, this is a descriptive paper with very limited data. The authors should not try to embellish it with irrelevant information. They should focus on the results and speculate as to why they might be significant in view of what is known for other systems.
Finally, I insist that a personal opinion cannot be taken as a scientific validation (I am referring to the inclusion of a statement by a neuropathologist that the cells in question are oligodendrocytes.

Experimental design

No comments

Validity of the findings

This paper documents the presence of perineuronal cells in certain species of birds using histological techniques. The information might be the starting point of research into the structure and function of these cells with a more sophisticated technology.

Comments for the author

Writing a paper is like telling a story. It is important to keep the interest and attention of the reader. The sequence should be such as to generate a certain expectation as to what is next.
Keep in focus and do not add more information than strictly necessary.
Some paragraphs in the conclusion could be moved to the Introduction while others belong to the Discussion. Keep the paper short!

---

## Round 0.3 · accepted · Accept

Many thanks again, for your submission, and for your responsiveness to the reviewers' comments.